# Position: Embodied AI Requires a Privacy-Utility Trade-off

**Xiaoliang Fan**[1]  **Jiarui Chen**[1]  **Zhuodong Liu**[1]  **Ziqi Yang**[1]  **Peixuan Xu**[1]  **Ruimin Shen**[1]  **Junhui Liu**[1]
**Jianzhong Qi**[2]  **Cheng Wang**[1]

## Abstract

Embodied AI (EAI) systems are rapidly transitioning from simulations into real-world domestic and other sensitive environments. However, recent EAI solutions have largely demonstrated advancements within *isolated stages* such as instruction, perception, planning and interaction, without considering their coupled privacy implications in high-frequency deployments where privacy leakage is often irreversible. This position paper argues that optimizing these components independently creates a systemic privacy crisis when deployed in sensitive settings, thereby advancing the position that privacy in EAI is a life cycle-level architectural constraint rather than a stage-local feature. To address these challenges, we propose Secure Privacy Integration in Next-generation Embodied AI (**SPINE**), a unified privacy-aware framework that treats privacy as a dynamic control signal governing *cross-stage* coupling throughout the entire EAI life cycle. SPINE decomposes the EAI pipeline into various stages and establishes a multi-criterion privacy classification matrix to orchestrate contextual sensitivity across stage boundaries. We conduct preliminary simulation and real-world case studies to conceptually validate how privacy constraints propagate downstream to reshape system behavior, illustrating the insufficiency of fragmented privacy patches and motivating future research directions into secure yet functional embodied AI systems.

---

[1] Fujian Key Laboratory of Urban Intelligent Sensing and Computing, School of Informatics, Xiamen University, Xiamen 361005, P. R. China  [2] School of Computing and Information Systems, The University of Melbourne, Australia . Correspondence to: Xiaoliang Fan <fanxiaoliang@xmu.edu.cn>.

*Proceedings of the 43$^{rd}$ International Conference on Machine Learning*, Seoul, South Korea. PMLR 306, 2026. Copyright 2026 by the author(s).

## 1. Introduction

Embodied Artificial Intelligence (EAI) systems are rapidly transitioning from laboratory benchmarks to real-world deployment in domestic and industrial environments (Duan et al., 2022; Liang et al., 2025; Ma et al., 2025; Sapkota et al., 2025; Kawaharazuka et al., 2025; Feng et al., 2025; Tan et al., 2025; Neupane et al., 2024). This shift highlights a critical, yet unresolved question: *How to build embodied agents that are privacy-aware by design, rather than privacy-patched by afterthought?* While current solutions prioritize mostly task success rates within isolated stages such as instruction understanding, environment perception, action planning and physical interaction, privacy is often treated as a secondary concern rather than a core goal of the EAI system design. **This position paper argues for a paradigm shift within the EAI community—moving away from isolated, stage-specific patches towards a holistic, tradeoff-aware architecture designed to integrate privacy-utility optimization in the EAI life cycle.**

Existing EAI solutions have advanced within isolated stages—instruction, perception, planning, and interaction (Tan et al., 2025). They reveal a systemic gap where privacy cannot be secured by component-level interventions. First, privacy leakage is compositional across stages. Stage-local patches, such as visual obfuscation in environment perception (Kim et al., 2019; Choi et al., 2025), often fail to prevent sensitive information from re-emerging downstream through reconstructed SLAM maps or policy execution traces (Yang et al., 2025; Sullivan & Mutlu, 2025). Second, the privacy-utility trade-off acts as a non-linear safety constraint. Aggressive restrictions in action planning (Shome et al., 2023; Yu et al., 2024) do not merely degrade efficiency, but can destabilize system control, compounding into wrong behaviors or even physical collisions (Zhu et al., 2022; Yeke et al., 2025). Consequently, trustworthy EAI deployment requires a unified framework that propagates privacy constraints across stage boundaries while characterizing operational limits imposed by this trade-off.

Furthermore, the design of privacy-aware EAI systems is constrained not only by technical limitations, but also by multi-jurisdictional privacy regulations (European Parliament and Council of the European Union, 2016; Califor-

nia State Legislature, 2018; U.S. Congress, 1996; Standing Committee of the National People's Congress, 2021). While these legal frameworks offer high-level normative principles, they remain largely agnostic to the staged, closed-loop nature of embodied systems. As a result, they offer limited operational guidance on how privacy requirements should be enforced across EAI stages. This regulatory–technical gap complicates the translation of abstract legal mandates into concrete EAI system design. Addressing this gap calls for a hierarchical classification scheme that maps high-level legal constraints to stage-aware mitigation strategies throughout the EAI life cycle.

To address these challenges, we propose **SPINE** (Secure Privacy Integration in Next-generation Embodied AI), a unified framework that treats privacy not as a localized patch, but as a first-class, dynamic control signal governing the entire EAI life cycle. We contend that bridging the gap between privacy and utility requires a fundamental shift: moving beyond component-level interventions toward an architectural paradigm of *Embodied Privacy*.

Specifically, a systematic framework must resolve the following interconnected questions: a) how to maintain rigorous cross-stage consistency to ensure that localized privacy interventions do not propagate through the system to trigger cascading utility failures; b) how to orchestrate a dynamic classification mechanism that adaptively balances the granularity of data sensitivity against the constraints of real-time computational overhead; c) how to effectively counteract the temporal accumulation of data leakage that emerges over long-horizon and complex multi-step interactions; d) how to seamlessly synthesize disparate privacy primitives into a cohesive, reliable control loop without compromising overall system stability or architectural integrity; and e) how to formally benchmark the "cost of privacy" by precisely quantifying its empirical impact on mission-critical metrics, such as task success rates and path-planning efficiency.

**In response**, SPINE introduces a multi-criterion privacy classification matrix (i.e., from L1 to L4) designed to orchestrate contextual sensitivity across stage boundaries. SPINE provides the first quantitative insights into how privacy constraints reshape agent behavior in EAI systems and defines operational boundaries of the privacy-utility trade-off.

## 2. Why Privacy Matters in Embodied AI

### 2.1. Privacy as a Cohesive Life cycle Property

Privacy matters in Embodied AI not as a feature of individual components, but as a cohesive property in the life cycle. A typical EAI pipeline spans instruction understanding, environment perception, action planning, and physical interaction (Tan et al., 2025), involving the iterative transformation of information at each stage. As a result, privacy

constraints must be maintained continuously across these transitions to ensure system integrity. We argue that current localized "patches" (Tan et al., 2025) are fundamentally insufficient because sensitive data is not static, but evolves throughout the EAI life cycle.

For instance, a robot might successfully anonymize faces during *perception*, yet its *planning* logs could still leak a user's specific medical condition—such as inferring Parkinson's disease by recording the frequent, precise movements required to retrieve anti-tremor medication. Similarly, a harmless *instruction* such as "organize the desk" may inadvertently reveal the existence of confidential financial documents when combined with *perception* data derived from an RGB-D camera. Consequently, EAI systems urgently require a cohesive, life cycle-wide privacy architecture to prevent fragmented leaks across stages from being aggregated into sensitive contexts and causing irreversible breaches.

### 2.2. Privacy as a Variable of System Utility

We further emphasize that privacy in EAI is a dynamic variable that defines the agent's operational boundaries. Balancing between privacy and utility is difficult, as gains in protection often come at the expense of performance (Butler et al., 2015). For instance, in the perception stage, excessively masking environmental inputs to hide sensitive background details might strip away critical semantic cues needed to identify a specific target object, directly leading to task failure. Conversely, a robot that exclusively prioritizes task success may capture environmental data far beyond task requirements. For instance, scanning an entire Bedroom to locate a single object might unintentionally expose sensitive personal items that are completely irrelevant to the task.

Crucially, privacy requirements are not static but vary by context. Current fixed privacy policies in the EAI community (Miao et al., 2025) may function in controlled labs, yet they often fail during real-world deployment (Zhou & Wang, 2022). This lack of adaptability creates a major barrier to task generalization, as an embodied agent cannot adjust its privacy-utility balance when moving between different environments. Consequently, an EAI-specific classification mechanism is essential to prevent this rigid approach from compromising both the agent's practical utility and its reliability across various settings.

## 3. Position: Navigating EAI Privacy-Utility Trade-off via the SPINE Framework

We contend that the structural vulnerabilities inherent in current EAI systems cannot be mitigated by fragmented privacy patches. Instead, we propose **SPINE**, a unified framework that treats privacy as a dynamic control signal

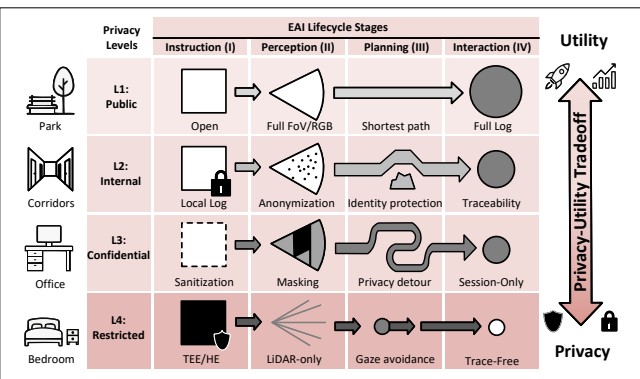

*Figure 1.* Conceptual architecture of SPINE maps the evolution of technical primitives across the four EAI life cycle stages (columns) against the four privacy levels (rows). The vertical gradient illustrates a strategic shift from utility-first environments at L1 (Park) to privacy-critical scenarios at L4 (Bedroom).

governing cross-stage coupling in the EAI life cycle. First, we establish a multi-criterion privacy classification matrix that explicitly categorizes embodied tasks into L1 to L4. Second, we present the conceptual architecture of SPINE. Third, we elaborate on the adaptive privacy orchestration across the EAI stages. Finally, we characterize operational boundaries of privacy–utility trade-off.

### 3.1. Multi-criterion Privacy Classification Matrix

To operationalize privacy across the EAI life cycle, we introduce a unified four-level privacy classification (Table 1) that transcends the conventional binary distinction between "public" and "private" data. This taxonomy functions as the architectural grammar for SPINE, categorizing embodied interactions into four tiers: L1 (Public), L2 (Internal), L3 (Confidential), and L4 (Restricted), based on their EAI scenario and privacy–utility trade-off priority. Importantly, the scenarios are selected to reflect high-frequency real-world deployments where privacy leakage is irreversible and tightly coupled with safety or regulatory cost. As detailed in Table 1, each level dictates specific operational privacy-aware techniques, spanning from universal API interfaces and federated learning for L1/L2, to multi-party computation (MPC) and homomorphic encryption for L3/L4. By instituting these rigorous definitions, the agent can dynamically orchestrate the privacy–utility trade-off, invoking stringent technical primitives only when necessitated by contextual sensitivity, thereby maintaining operational utility without compromising privacy mandates.

### 3.2. Conceptual Architecture of SPINE

To illustrate how privacy classifications instantiate concrete architectural decisions, Figure 1 presents the conceptual architecture of the SPINE framework. The diagram spec-

ifies the technical primitives required across the four EAI stages: Instruction Understanding, Environment Perception, Action Planning, and Physical Interaction. Crucially, this visualization synthesizes our core methodology by demonstrating how privacy as a dynamic control signal spans the entire pipeline to prevent cross-stage privacy leakage (to be detailed in Section 3.3). Furthermore, it explicitly maps the privacy–utility trade-off, visualizing the transition from utility-prioritized public zones in L1 (e.g., park) to rigorously restricted zones in L4 (e.g., Bedroom), to be detailed in Section 3.4.

### 3.3. Adaptive Privacy Orchestration: From Public Utility to Restricted Safety

Rather than applying a monolithic privacy policy, SPINE functions as a dynamic orchestration that adjusts the agent's behavior based on the privacy classification matrix defined in Table 1. Crucially, this tiered approach ensures that privacy constraints are enforced holistically across the life cycle, spanning instruction, perception, action, and interaction, thereby preventing systemic leakage caused by treating these stages as isolated, fragmented components.

In **Level 1 (Public)** scenarios, such as navigating a park, SPINE adopts a utility-maximizing configuration to ensure efficiency parity with state-of-the-art generalist agents. For the instruction understanding stage, user commands are routed directly to cloud-based Large Language Models (LLMs) to leverage full reasoning capabilities without semantic masking. Hence, the environment perception stage operates with unrestricted high-fidelity RGB-D streams, allowing the agent to utilize open-vocabulary vision models for precise object detection. In the action planning stage, the navigation algorithm optimizes strictly for the shortest path, treating the environment as a purely geometric space without privacy constraints. Finally, during physical interaction, execution logs including sensor data are retained to facilitate error recovery and system debugging, prioritizing robust performance over data minimization.

As the context shifts to **Level 2 (Internal)** settings, e.g., office corridors, the architecture tightens to protect bystander anonymity while maintaining workflow continuity. The instruction understanding stage remains largely standard but introduces local logging for traceability. The primary shift occurs in the environment perception stage, where SPINE activates affordance-based filtering. While depth channels remain active for obstacle avoidance, RGB streams undergo real-time anonymization to mask identity markers such as faces or license plates before data enters the visual backbone. Consequently, the action planning stage generates trajectories based on this depersonalized semantic map, ensuring that navigation geometry is preserved while biometric privacy is secured. The physical interaction stage contin-

*Table 1.* Multi-criterion privacy classification matrix in SPINE.

| Level | EAI Scenario | Privacy-Utility Trade-off Priority | Privacy-aware Techniques | Legal Provisions |
|---|---|---|---|---|
| L1 Public | Crowd navigation in a park; Operating service robots in airports | Utility First: Minimal data restrictions to ensure navigation efficiency | Public pre-trained models; Universal API interfaces | Not subject to the GDPR |
| L2 Internal | Hotel cleaning robots in shared corridors; Office delivery robots handling employee logistics | Balanced: Environmental layouts are mapped but specific identity markers are blurred | Data Desensitization and Anonymization; Trusted Execution Environment; Private Set Intersection; Federated Learning | GDPR Recital. 26 (Scope of Application / Data Threshold) |
| L3 Confidential | Office assistant robots operating in private office; In-room service robots in hotels | Privacy-Leaning: High protection for home details and local processing encouraged | Secure Multi-Party Computation; Differential Privacy; Federated Learning | GDPR Art. 4 (General Personal Data); GDPR Art. 35 (DPIA) |
| L4 Restricted | Elder-care assistance in Bedroom (fall detection); Patient monitoring in hospital wards | Privacy First: Maximum constraints (e.g., volatile memory); Utility may be limited to essential safety | Zero-Knowledge Proof; Fully Homomorphic Encryption; Cutoff the visual feeds | GDPR Art. 9 (Sensitive Data); GDPR Art. 10 (Judicial / Criminal Data); GDPR Art. 35 (DPIA) |

ues to operate with standard latency, but visual logs have identifiable features removed before storage.

Moving into **Level 3 (Confidential)** environments, such as a private office room, SPINE enforces a decoupling of sensitive intent and data. In the instruction understanding stage, the system activates a context-aware verification layer. Instructions are pre-processed locally to sanitize sensitive entities, such as replacing specific medication names with abstract object classes. The environment perception stage restricts the field of view, employing dynamic masking to block out non-task-relevant background details like documents on a table. In action planning, the system introduces a privacy cost map, where private zones are assigned high traversal penalties, forcing the planner to generate trajectories that physically circumvent sensitive areas unless necessary. For physical interaction, data persistence is restricted, and execution logs are stored only for the duration of the session and are encrypted at rest, balancing the need for immediate feedback with long-term privacy protection.

Finally, in **Level 4 (Restricted)** scenarios, such as assisting in a Bathroom, SPINE reconfigures the entire pipeline into a safety-critical isolation mode. The instruction understanding stage serves as a strict firewall, commands are processed exclusively within a local Trusted Execution Environment, preventing any raw audio or semantic intent from leaving the device. Crucially, the environment perception stage enforces source-level restriction across modalities. For instance, the system might terminate the task, or trigger a hardware-level cutoff of visual feeds, forcing the agent to rely solely on privacy-preserving modalities, e.g., LiDAR or sparse depth maps, to prevent visual reconstruction. The action planning stage shifts to a privacy-compliant primitive, where

the agent not only avoids unnecessary movement but also actively orients sensors away from sensitive sub-regions like beds during operation. In the final physical interaction stage, SPINE mandates in-memory-only execution. All control loops run in volatile memory with a secure wipe protocol triggered immediately upon task completion, while external status LEDs provide legible signaling of this privacy mode to the user. This configuration accepts a reduction in multi-modal capability to provide a verifiable guarantee that what happens in the room, stays in the moment.

### 3.4. Privacy-Utility Trade-off: From Simulation to Deployment

Standard EAI frameworks often rely on static privacy rules that may suffice in controlled simulations but fail to adapt to the fluidity of real-world deployment. In contrast, SPINE serves as an adaptive regulation mechanism that dynamically modulates the agent's behavior according to the sensitivity levels outlined in Table 1. This tiered approach ensures that the system maximizes utility in low-risk scenarios while enforcing rigid architectural constraints in high-stakes environments, providing a scalable path for transitioning embodied agents from simulated training to physical deployment. We describe how SPINE could optimize the privacy-utility trade-off at each level of the Privacy Classification Matrix (Table 1) as follows.

In **Level 1 (Public)** scenarios, SPINE leverages its adaptive nature to prioritize a "Utility First" strategy. Since these scenarios typically do not involve sensitive personal data subject to strict GDPR constraints, the framework allows the agent to operate with minimal data restrictions. By utilizing public pre-trained models and universal API interfaces,

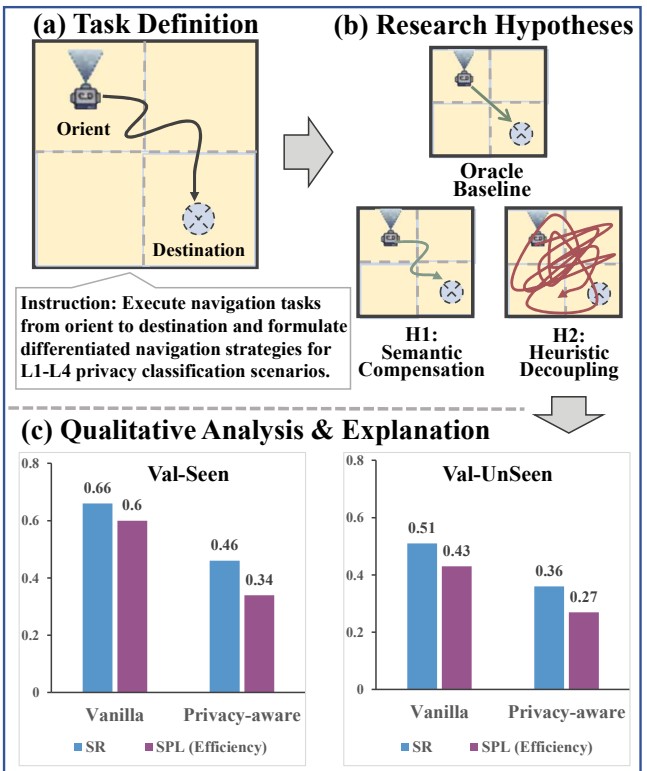

*Figure 2.* Embodied navigation case study under SPINE framework: (a) Task Definition: long-range planning from orient to destination; (b) Research Hypotheses: visualization of H1 (Semantic Compensation) and H2 (Heuristic Decoupling); (c) Qualitative Analysis of H1 and H2 through $SR$ and $SPL$ metrics.

SPINE maximizes navigation efficiency and robustness, ensuring that in low-risk settings, the agent achieves high operational performance comparable to controlled simulations without being hindered by unnecessary privacy overhead.

Moving to **Level 2 (Internal)** contexts, SPINE shifts to a "Balanced" trade-off that preserves operational utility while introducing necessary identity protections. The framework dynamically modulates behavior to map environmental layouts for tasks such as cleaning, while simultaneously activating data desensitization and anonymization techniques to blur specific identity markers. By incorporating Trusted Execution Environments and Federated Learning, SPINE maintains the system's ability to navigate and function effectively in shared spaces while adhering to data thresholds outlined in GDPR Recital 26.

For **Level 3 (Confidential)** settings, SPINE implements a "Privacy-Leaning" priority where the protection of personal habits supersedes maximum efficiency. The framework enforces a shift towards local processing and employs advanced privacy-preserving techniques such as Secure Multi-Party Computation and Differential Privacy. This trade-off ensures that while the agent retains the capability to per-

form tasks like organizing items, the granularity of data exposure is strictly controlled to safeguard general personal data (GDPR Art. 4), accepting a calculated reduction in cloud-dependent utility to ensure confidentiality.

Finally, in **Level 4 (Restricted)** environments, SPINE enforces a "Privacy First" paradigm with rigid architectural constraints. In these high-stake scenarios, the framework limits utility strictly to essential safety functions, such as fall detection, while mandating the use of volatile memory, Zero-Knowledge Proofs, and Fully Homomorphic Encryption. By prioritizing the protection of sensitive and judicial data (GDPR Art. 9 & 10) over general capability, SPINE provides a scalable path for deployment in the most vulnerable spaces, ensuring that physical safety does not come at the cost of fundamental privacy rights.

## 4. Case Study

To conceptually validate the technical feasibility of our position, we conduct a focused case study on embodied navigation under the SPINE framework. Rather than introducing a new navigation algorithm or competing with state-of-the-art methods, our objective is to empirically expose how stage-local privacy interventions propagate through the embodied life cycle, affecting downstream planning efficiency. As shown in Figure 2, the case study consists of three components. First, *task definition* (Figure 2 (a)) centers on a long-range navigation task where an agent transitions from low-sensitivity public zones to high-sensitivity restricted areas under diverse privacy constraints. Within this setting, we introduce isolated privacy patches into a vanilla baseline, a privacy-free visual language navigation (VLN) system to examine how privacy interventions applied at the perception stage might propagate to downstream planning stage. Second, we formulate two *research hypotheses* (Figure 2(b)): (H1) Semantic Compensation, which posits that high-level semantic intent can partially mitigate the effects of perceptual degradation, and (H2) Heuristic Decoupling, which hypothesizes that severe perceptual loss fractures the coupling between perception and planning. Third, we evaluate these hypotheses through *qualitative analysis* of experiments conducted in both simulated and real-world environments. The empirical results, summarized in Figure 2(c), reveal that while task success rates undergo a moderate decline, the degradation in planning efficiency is significantly more pronounced. This divergence quantitatively characterizes the privacy–utility trade-off, demonstrating that privacy compliance in embodied navigation can come at the expense of execution efficiency.

### 4.1. Task Definition

Figure 2 (a) presents our controlled navigation setup for quantifying the privacy-utility trade-off. In this "long-range

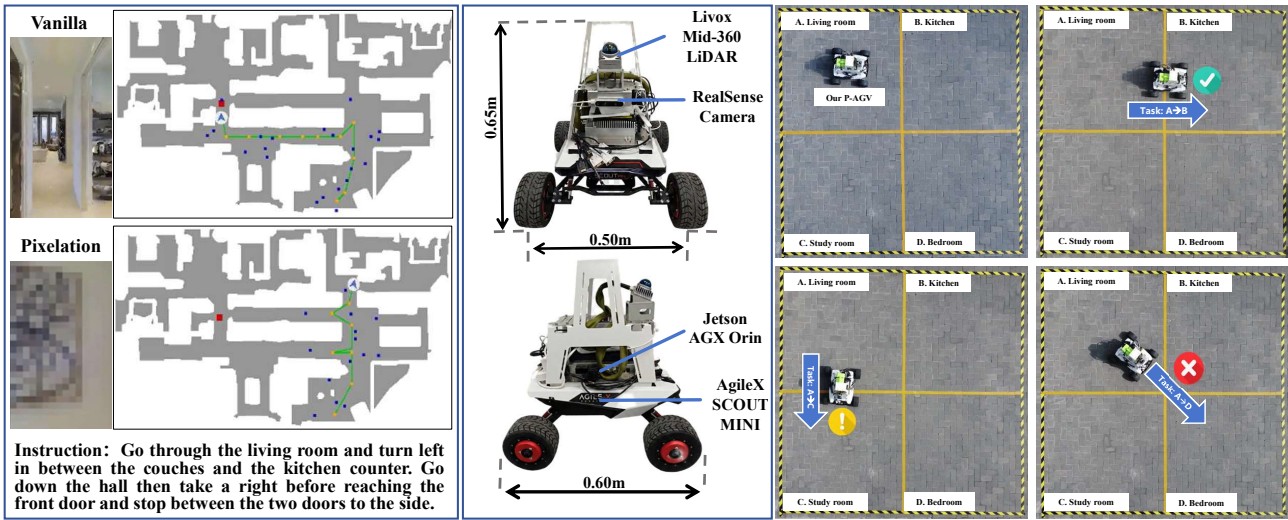

*Figure 3.* Experimental settings and results: simulated environment (left), AGV platform (middle), and real-world experiments (right).

navigation" task, the agent navigates from a Public Zone (L1) to a Restricted Zone (L4), providing a benchmark to test spatial reasoning under stringent privacy constraints.

**In the simulation experiments**, we evaluate the *R2R-CE* task using the *Habitat* simulator by the ETPNav algorithm (An et al., 2024). To quantify the navigation performance, we adopt two benchmark metrics: success rate ($SR$) and success weighted by path length ($SPL$). To emulate privacy-constrained environments, we introduce perceptual-stage interventions by applying pixelation patches to the RGB input (Figure 3, left). This setup effectively obscures critical visual anchors, forcing the embodied agent to rely on residual information while executing semantic instructions.

**In the real-world experiments**, we deploy an automatic guided vehicle (AGV) built upon an AgileX SCOUT MINI mobile hardware platform (Figure 3, middle). The AGV is equipped with an integrated sensing and computing suite, including a Livox Mid-360 LiDAR for high-precision spatial mapping, a RealSense Camera for visual input, and a Jetson AGX Orin module for real-time inference and navigation control. Utilizing the aforementioned hardware, we conduct privacy-aware experiments in a 4m × 4m physical testbed (Figure 3, right) subdivided into quadrants A through D. These areas represent domestic environments (e.g., Bedroom or Bathroom), each governed by a privacy level ranging from L1 (Public) to L4 (Restricted) as detailed in Section 3.1. The agent's objective is to complete navigation tasks, such as "navigates between rooms for elderly fall detection" by dynamically adapting its behavior to the specific privacy constraints of each zone.

### 4.2. Research Hypotheses

Based on the SPINE framework (Section 3.2), we formulate two research hypotheses to predict how privacy constraints alter the internal mechanisms of an embodied agent's planning architecture. We emphasize that these hypotheses are not tied to a specific algorithm, but capture structural effects induced by privacy constraints under the SPINE abstraction.

First, we propose H1 (Figure 2 (b)): Semantic Compensation. This hypothesis posits that the EAI life cycle possesses an inherent "semantic persistence". Specifically, H1 asserts that even when visual data in Stage II (Environment perception) is sanitized to protect privacy, the high-level goals derived from Stage I (Instruction understanding) continue to act as a strong semantic anchor. This allows the agent to maintain a basic task success rate ($SR$) because the "meaning" of the task effectively travels across stage boundaries, compensating for the limited visual inputs.

Second, we propose H2 (Figure 2 (b)): Heuristic Decoupling. This hypothesis addresses the performance divergence observed in restricted zones. H2 states that at a high level (L4), the loss of visual details precipitates a "logic break" between perception and planning. Without clear visual landmarks, the agent's path-planning in Stage III (Action planning) loses its grounding ability. Consequently, the navigation policy is forced to regress from efficient heuristics into simple, stochastic search patterns (such as a random walk), leading to a significant decline in navigation efficiency ($SPL$).

### 4.3. Qualitative Analysis and Explanation

**Results of simulation**. Experimental results demonstrate a distinct divergence in performance metrics across var-

ied environments, offering empirical substantiation for the mechanisms hypothesized in H1 and H2. This trend remains robust across both Val-Seen and Val-UnSeen datasets. Specifically, in the Val-Seen scenario (Figure 2 (c) Left), the Success Rate ($SR$) dropped from 0.66 to 0.46 (a 30.3% reduction), whereas the Success weighted by Path Length ($SPL$) underwent a substantial reduction from 0.60 to 0.34 (43.3%). A comparable disparity persists in the Val-UnSeen environment (Figure 2 (c) Right), where the reduction in $SPL$ ($-37.2\%$) notably outpaces that of $SR$ ($-29.4\%$).

These non-linear variations provide empirical support for H1: Semantic Compensation, suggesting that high-level intent anchoring from instruction stage preserves baseline task success despite constrained visual inputs, thus averting systemic failure. Furthermore, the observation that $SPL$ degradation significantly exceeds $SR$ loss indicates a compromised capacity for coherent path planning, forcing the agent toward exploratory trajectories. This validates H2: Heuristic Decoupling, as privacy-induced sanitization of visual landmarks in perception stage disrupts the synergy between navigation policies and efficient planning.

**Results of real-world experiments**. As illustrated in Figure 3 (right), we configured our AGV with a privacy-aware embodied navigation algorithm, initialized in the "Living Room"—an area designated as L2 (Internal). In this state, the agent is authorized to utilize its camera for perception. For the navigation task A → B (Living Room to Kitchen), as the Kitchen is also categorized as a L2 zone, the AGV proceeds directly without additional privacy interventions. In contrast, the task A → C (Living Room to Study Room) necessitates a transition to an L3 (Confidential) zone. Consequently, the agent is mandated to implement pre-defined privacy preservation measures—such as image pixelation or camera gimbal depression—prior to entry. Finally, for task A → D (Living Room to Bedroom), navigation is strictly prohibited as the Bedroom is classified as an L4 (Restricted) zone. These empirical results demonstrate that our SPINE framework effectively enforces the proposed privacy classification, executing adaptive navigation strategies tailored to the sensitivity of diverse functional regions.

## 5. Alternative Views

Diverse perspectives exist regarding the architectural priorities of our framework, particularly concerning the tension between rapid innovation and privacy constraints. First, one thought argues that given EAI's birth, research should prioritize agent capabilities above all else. There is another concern (Huang et al., 2025) that overly restrictive privacy protocols may handicap the scaling potential of embodied AI, which inherently demands massive computational resources for performance optimization. Alternatively, some researchers advocate for privacy behaviors to be learned

implicitly (Li et al., 2024), rather than enforced through structural modularity. From this viewpoint, imposing rigid boundaries across the EAI pipeline might compromise the inherent synergies of end-to-end learning paradigms. Third, many users prefer the seamless, low-latency experience of cloud-integrated EAI systems, which often surpasses privacy-aware edge solutions that are frequently hindered by local resource constraints. However, we argue that the physical embodiment of EAI systems inevitably raises the risks. Unlike purely digital software, EAI agents interact with the physical world, posing unique and irreversible risks to human safety and spatial privacy. This reality requires a paradigm shift from probabilistic alignment, which relies on the hope that a model will "behave", to verifiable architectural guarantees. In this context, the pursuit of operational efficiency must not compromise the fundamental requirement of secure, reliable EAI deployment.

## 6. Related Works

### 6.1. Privacy in Embodied AI

Privacy risks in EAI emerge as a cumulative effect across the entire instruction-to-interaction life cycle (Tan et al., 2025). In the instruction understanding stage, recent works focus on gating user requests through privacy-preserving language representations, secrecy evaluation via contextual integrity, and task-oriented instruction rewriting to decouple sensitive content from downstream policies (Qu et al., 2021; Mireshghallah et al., 2024; Wang et al., 2025; Mishra et al., 2025). During perception, many works emphasize representation-level sanitization (e.g., low-resolution anonymization), real-time filtering, and trade-offs in teleoperation, alongside alternative modalities like LiDAR (Kim et al., 2019; Choi et al., 2025; Butler et al., 2015; Huang et al., 2025; Amatare et al., 2024; Huang et al., 2026; Tsaprazlis et al., 2025). Even with sanitized perception, action planning remains vulnerable, as trajectories may leak routines or intent. This motivates privacy-aware motion planning, federated VLN, and secure multi-robot coordination (Shome et al., 2023; Savkin et al., 2025; Zhou & Wang, 2022; Li et al., 2019; 2024). Finally, during physical interaction, privacy hinges on disclosure behaviors and long-horizon data retention. Current efforts mainly address adaptive sequential decision-making and privacy-recognition benchmarks for social robots (Taherisadr et al., 2023; Dietrich et al., 2023; Sullivan et al., 2025; Yang et al., 2022).

In summary, current privacy-aware EAI solutions remain largely stage-local and disjoint. Instead, our position shifts the perspective by treating privacy not as stage-specific patches, but as a dynamic control signal that governs the entire EAI life cycle through cross-stage coordination.

## 6.2. Privacy-Utility Trade-offs

The inherent tension between data minimization and system performance creates a fundamental trade-off that manifests across diverse intelligent systems. In control theory, privacy is often the "cost" of deviating from energy-efficient trajectories (Savkin et al., 2025). In multimedia and internet of things, many works optimize real-time data granularity under resource constraints (Topalli et al., 2025; Kil et al., 2024). Recommender Systems also face a trilemma among differential privacy, accuracy, and fairness (Parsarad & Wagner, 2025). Generative AI balances model expressiveness against the risk of leaking sensitive visual attributes (Tsaprazlis et al., 2025; Mireshghallah et al., 2024). Specific to embodied systems, aggressive privacy filtering (e.g., blurring or resolution reduction) directly reduces situational awareness and navigation success (Kim et al., 2019; Butler et al., 2015; Huang et al., 2025). Furthermore, a utility ceiling exists between the reasoning power of cloud models and the data ownership of local execution. While federated learning attempts to bridge this gap, it struggles with communication overhead and security vulnerabilities in dynamic environments (Zhou & Wang, 2022; Zhang et al., 2024).

Unlike existing approaches that enforce rigid, binary compromises between local security and cloud utility, we propose an architectural paradigm that treats privacy as a dynamic control signal. This allows SPINE to orchestrate the privacy–utility trade-off across the entire embodied life cycle, modulating constraints based on real-time contextual sensitivity rather than fixed penalties.

## 7. Call to Action and Future Directions

While the Embodied AI landscape is expanding rapidly, its development remains structurally fragmented. We argue that building secure EAI systems requires a shift from isolated component-level optimization toward integrated frameworks in which privacy serves as a holistic, tradeoff-aware design primitive. Building on the insights of the SPINE framework, we outline the following coordinated actions for diverse stakeholders (e.g., system and platform providers, researchers and benchmark developers, device manufacturers, hardware and software providers) as well as strategic and future research directions for the Embodied AI community.

**Verifiable Cross-Stage Privacy Consistency.** *System and platform providers* must prioritize a paradigm shifting from localized patches to the cross-stage protocols that maintain consistent policies from high-level instruction understanding to downstream physical interaction. The core challenge lies in providing verifiable interfaces that ensure privacy requirements, such as those classified as L4 (Restricted), impose traceable constraints upheld from perception-level data

redaction to trajectory masking. Future development should focus on formal verification and unified communication protocols that allow for the auditing of cross-module invariants to prevent downstream reconstruction of task-irrelevant information.

**Auditable Privacy–Utility Co-optimization.** *Researchers and benchmark developers* should establish standardized frameworks that treat the privacy–utility balance as a multi-objective optimization problem. Rather than opposing cloud or end-to-end systems, we advocate for their usage under auditable constraints: low-risk (L1–L2) tasks may utilize cloud resources, while sensitive (L3–L4) operations must prioritize Edge or TEE-based execution (Liu et al., 2025). Future evaluation platforms must characterize measurable signals—such as information exposure, gradients, and data retention (Zhang et al., 2022)—to enable rigorous comparison of how privacy constraints reshape agent behavior across diverse threat models.

**Verified Context-Adaptive Privacy Orchestration.** *Deployment entities and device manufacturers* must implement adaptive orchestration that switches privacy labels based on an agent's physical, social, and temporal context. We argue for the adoption of explicit criteria where tasks (e.g., "fetching") escalate from L1 in commercial warehouses to L3 in private residences based on spatial ownership and bystander presence. Future field deployments should explore real-time orchestration that switches sensing modalities (e.g., RGB to LiDAR) using quantifiable risk signals, while automatically elevating protection when exposure exceeds predefined privacy budgets.

**Verifiable Infrastructure through Co-design.** Finally, *hardware and software providers* must emphasize co-design to support a multi-level privacy hierarchy with real-time enforcement. While federated learning enables privacy-aware Distributed encryption co-training (Zhou & Wang, 2022; Miao et al., 2025), integrating software efficiency with hardware acceleration, such as PEFT (Jabbour et al., 2025), is essential for achieving L4-level protection under strict constraints (Black et al., 2025). These stakeholders should collaborate to deliver verifiable interfaces and compliance mapping, ensuring that privacy guarantees are not theoretical but auditable in embodied systems.

## 8. Conclusion

The rapid and pervasive deployment of Embodied AI (EAI) across diverse physical environments has significantly intensified the inherent tension between operational utility and data privacy, exposing the critical structural limitations of existing fragmented approaches. In this position paper, we argue that next-generation EAI architectures must transcend isolated patches and evolve toward a holistic, tradeoff-aware

paradigm that integrates privacy as a core design principle. To this end, we introduce SPINE, a unified framework that treats privacy as a dynamic, programmable control signal governing complex cross-stage coupling throughout the entire EAI life cycle. Our empirical results demonstrate how these enforced privacy constraints fundamentally reshape agent behaviors and delineate the precise operational boundaries of the privacy–utility trade-off. Future work could formalize comprehensive threat models across diverse leakage channels—including raw sensor streams, latent embeddings, and gradients—to enable verifiable and auditable privacy constraints at critical EAI architectural interfaces.

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
