# OpenReview forum: "Position: Embodied AI Requires a Privacy-Utility Tradeoff"
_ICML.cc/2026/Position_Paper_Track — ICML 2026 Position Paper Track regular_

### Official Review · Reviewer_pNDA · 2026-03-04

**Significance:** 3
**Argument Clarity:** 2
**Rating:** 4
**Confidence:** 3

**Questions:**

1. What is the explicit threat model you target (adversary, capabilities, access to logs/maps/queries), and what privacy properties do you aim to guarantee or measure?

2. Can you provide an operational definition for each privacy level (L1–L4): measurable criteria, triggering rules, and how level transitions are enforced and audited?

3. Can you outline a minimal protocol (tasks, interventions, leakage metrics, utility metrics) that the community could adopt to evaluate “privacy-aware embodied behavior” consistently?

**Alternative Views Section:**

Yes

**Compliance With Llm Reviewing Policy A Conservative:**

Affirmed.

**Discussion Potential:**

2

**Final Justification:**

I appreciate the authors' effort in rebuttal and willing to slightly raise my score, however, the rebuttal cannot fully address my concern. For instance, authors give examples of potential adversary by "honest-but-curious cloud providers observing uploaded queries" and "compromised storage/insiders with offline access to retained logs, maps", which seems to be too hypothetic since embodied devices are often likely to be distributed across different service providers and different edge devices. Getting access to all the information across those distributed devices/cloud servers seems to be a strong assumption for the attacker which can launch a lifecycle-level privacy attack. Since I do not work on privacy, I am not very sure about how feasible could this privacy protection scheme work in real practice.

**Paper Summary:**

This position paper argues that as Emboddied AI moves from simulators to real-world environments, privacy must be treated as a lifecycle-level constraint rather than a stage-local patch. Authos claim that privacy leakage is compositional across stages, therefore privacy requirements must be propagated across the entire EAI pipeline. The paper proposed SPINE, a unified framework that treats privacy as a dynamic control signal.

**Position:**

Yes

**Position In Title:**

Yes

**Related Work:**

2

**Strengths And Weaknesses:**

Strength:

1. The paper is well-motivated: the motivation is practical in real-world scenarios. It highlights compositional privacy leakage across pipeline stages which lacks considerations in prior works.

2. Actionable starting point via privacy classification matrix and the associated toolbox which offers a concrete vocabulary for thinking about deployment requirements.

Weakness:

1. The L1-L4 classification seems to be too rough and may need further considerations. The paper does not provide crisp boundaries or triggering conditions. How transitions between levels could be difficult to handle and verify in real use.

2. Threat model and security objectives are insufficiently specified. The paper discusses privacy leakage through logs / maps / trajectories, but it does not clearly define the adversary (whether he is a cloud operator, insider or external attacker?) and their capabilities (offline access? online queries? side channels?) or the privacy goals (attribute inference ? identity protection?) Without an explicit threat model, it is difficult to evaluate privacy guaratees.

**Support:**

2

---

> ### Author Rebuttal · Authors · 2026-03-30
>
> We sincerely thanks for your valuable comments. We hope our response point-by-point below addresses your concerns.
>
> **Q1 Threat model and privacy properties to guarantee.**
>
> **Response**: We clarify that SPINE targets a lifecycle-level threat model where privacy leakage is compositional across instruction, perception, planning, and interaction. Concretely, we now explicitly define three adversaries: (i) honest-but-curious cloud/service providers accessing uploaded queries or intermediate data; (ii) compromised storage/insiders with offline access to retained logs, maps, or trajectories; and (iii) an external or over-privileged observer who may not access raw sensing directly but can still infer sensitive attributes.
>
> Protected properties scale with L1–L4 levels. These include identity protection, attribute-inference resistance, prevention of contextual reconstruction (from maps/traces), and bounded retention. L4 adds strict containment via local TEE, source-level sensing restrictions, and volatile execution.
>
> In the revised paper, We agree that the current manuscript does not state this threat model explicitly enough. In the revision, we will add a short paragr aph in Section 3.1/3.3 that defines the target adversaries, their access capabilities, the relevant leakage channels, and the main privacy properties protected by SPINE.
> Protected properties scale with L1–L4 levels, including identity protection, attribute-inference resistance, prevention of contextual reconstruction (from maps/traces), and bounded retention. L4 adds strict containment via local TEE, source-level sensing restrictions, and volatile execution.
>
> In the revised manuscript, we will add a paragraph in Sections 3.1 and 3.3 defining the target adversaries, their access capabilities, relevant leakage channels, and the primary privacy properties protected by SPINE.
>
> **Q2 Provide definitions of L1-L4.**
>
> **Response**: We agree that our SPINE needs to explicitly define the "how" regarding operational boundaries. We will add this in the revised mansucript, which could truns Table 1 from illustrative to operational.
>
> Specifically, we explicitly define each level (L1–L4) using a unified tuple $P_L = \{S, I, C, \Phi\}$—where $S$ is Scenario, $I$ is Information Flow, $C$ is Control Primitive, and $\Phi$ is Utility Objective—we turn these representative anchors into formal states. Here is the condensed operational logic for L1–L4, bridged by the pixelation strength ($K$) used in our case study.
>
> * **L1: Public (Utility-Maximizing):** Unrestricted data flow ($I \in \text{Cloud}$) to leverage maximum reasoning power. Logic: $K=0$ (Vanilla baseline); Privacy Cost $\approx 0$.
> $P_{L1} \rightarrow \lbrace S_{pub}, I_{cloud}, C_{null}, \Phi_{max} \rbrace $
>
> * **L2: Internal (Identity-Anonymized):** Hybrid flow focused on removing biometric identifiers ($B$) while preserving spatial geometry. Logic: $K$ is minimal; filters faces/plates but keeps navigation cues intact.
> $P_{L2} \rightarrow \lbrace S_{shared}, I_{hybrid}, C_{f(x)-B}, \Phi_{efficiency} \rbrace $
>
> * **L3: Confidential (Intent-Decoupled):** Local processing to sanitize semantic entities and enforce privacy-aware detours. Logic: $K > 0$; aggressive pixelation reduces visual fidelity to protect environmental context.
> $P_{L3} \rightarrow \lbrace S_{private}, I_{local}, C_{sanitise}, \Phi_{safety\_{first}} \rbrace $
>
> * **L4: Restricted (Zero-Knowledge/Isolation):** Stateless, verifiable isolation mode using non-reconstructable modalities (e.g., LiDAR). Logic: $K \rightarrow \text{Saturation}$; camera cutoff or task prohibition to ensure "what happens in the room, stays in the moment".
> $P_{L4} \rightarrow \lbrace S_{sensitive}, I_{isolate}, C_{FHE/TEE}, \Phi_{min\_{viable}} \rbrace $
>
> **Q3 Outline minimal protocol that could evaluate privacy-aware EAI.**
>
> **Response**: We propose a four-pillar protocol to standardize the evaluation of privacy-aware embodied behavior: (1) task families, e.g., navigation, manipulation, and social assistance; (2) lifecycle interventions, e.g., stage-wise controls spanning instruction sanitization, perception anonymization, privacy-aware planning, and data retention; (3) dual metrics, e.g., balanced reporting of utility (success, efficiency, safety) and leakage (re-identification or attribute-inference from maps/logs).
>
> Our navigation study serves as a concrete instantiation of this template, using pixelation strength ($K$) as a controllable privacy proxy and SR/SPL as utility metrics shown in Figure 2. We will revise the manuscript to explicitly frame our current experiments as an initial step toward this broader benchmark.

---

> > ### Author Rebuttal · Reviewer_pNDA · 2026-04-06
> >
> > Thanks for the rebuttal. To me, the threat model (adversary) needs to be defined more carefully and in details - right now, the threat model still seems to be quite vague hypothetic. For example, in real life, the embodied ai service can often be provided by different cloud /service providers; their offline logs /maps/ trajectories are often distributed among different devices. It will be preferable if the authors can propose real-life examples of such lifecycle-level privacy threat. The triggering conditions between L1-L4 scenarios are not answered, but I think it needs to be delibrately discussed in the paper. I also agree with Reivewer pTTZ that the paper seems to propose more a solution than a position.

---

### Official Review · Reviewer_2Gug · 2026-03-05

**Significance:** 3
**Argument Clarity:** 3
**Rating:** 4
**Confidence:** 3

**Questions:**

Please refer to Weaknesses.

**Alternative Views Section:**

Yes

**Compliance With Llm Reviewing Policy A Conservative:**

Affirmed.

**Discussion Potential:**

3

**Paper Summary:**

This position paper addresses a critical and underexplored challenge in Embodied AI (EAI): the inherent tension between operational utility and privacy preservation in real-world deployments. The authors argue that current EAI solutions rely on fragmented, stage-specific privacy "patches" (e.g., visual obfuscation in perception) that fail to account for cross-stage privacy leakage and non-linear trade-offs with system utility. To address this gap, they propose SPINE (Secure Privacy Integration in Next-generation Embodied AI), a unified framework that treats privacy as a dynamic control signal governing the entire EAI life cycle—spanning instruction understanding, environment perception, action planning, and physical interaction. SPINE introduces a four-level privacy classification matrix (L1-L4) tailored to EAI scenarios, mapping contextual sensitivity to stage-specific privacy-preserving techniques and regulatory compliance. Preliminary simulation and real-world case studies on embodied navigation validate the framework’s core hypotheses, demonstrating how privacy constraints reshape agent behavior and quantifying the privacy-utility trade-off (e.g., moderate declines in task success rate but pronounced drops in planning efficiency). The paper concludes with a call to action for stakeholders to prioritize cross-stage consistency, auditable co-optimization, and context-adaptive privacy orchestration in future EAI systems.

**Position:**

Yes

**Position In Title:**

Yes

**Related Work:**

3

**Strengths And Weaknesses:**

Strengths

1. The paper addresses a pressing gap in EAI research, where the rapid transition from simulated to real-world (e.g., domestic, healthcare) deployments has outpaced privacy considerations. By framing privacy as a life cycle-level architectural constraint rather than a local feature, it challenges the dominant paradigm of isolated privacy interventions and aligns with growing regulatory and ethical demands for trustworthy AI.
2.  SPINE’s four-level privacy classification matrix is a standout contribution—its integration of EAI-specific scenarios, privacy-utility priorities, technical primitives, and regulatory provisions (e.g., GDPR) provides a actionable roadmap for translating abstract privacy mandates into concrete system design. This structured approach fills a critical void between high-level policy and low-level implementation.
3. Unlike purely theoretical position papers, the authors support their arguments with a focused case study on embodied navigation, including both simulation (Habitat simulator) and real-world (AGV platform) experiments. The validation of hypotheses (Semantic Compensation and Heuristic Decoupling) quantifies the non-linear nature of the privacy-utility trade-off, adding rigor to the position and demonstrating practical implications.
4. The paper goes beyond proposing a framework to outline coordinated actions for diverse stakeholders (system providers, researchers, manufacturers, hardware/software developers). This broad perspective enhances the work’s impact, positioning it as a catalyst for community-wide change rather than a narrow technical contribution.
5. The authors effectively situate their position within existing literature on EAI privacy and privacy-utility trade-offs, clearly articulating how SPINE differs from fragmented, stage-local solutions and rigid binary trade-off approaches. The discussion of alternative views (e.g., prioritizing capability over privacy, implicit privacy learning) further strengthens the paper’s credibility by engaging with counterarguments.

Weaknesses

1. The empirical validation is restricted to embodied navigation tasks, with a narrow focus on visual perception constraints (e.g., pixelation patches). This limits the ability to generalize SPINE’s effectiveness to other EAI tasks (e.g., manipulation, social interaction) or other privacy leakage channels (e.g., audio data, trajectory patterns, latent embeddings in language models).
2. While SPINE’s privacy classification matrix is well-defined, the paper provides insufficient detail on how the framework dynamically adapts privacy levels in real time. For example, the mechanisms for detecting contextual shifts (e.g., transitioning from a public corridor to a private office) or resolving conflicts between privacy priorities and safety-critical requirements (e.g., fall detection in L4 restricted environments) are not formally specified.
3. The paper proposes strict L4 (Restricted) privacy measures such as "hardware-level cutoff of visual feeds" and "in-memory-only execution," but provides little discussion of their technical feasibility or performance overhead in resource-constrained EAI platforms (e.g., edge devices, low-cost robots). Practical challenges like latency, computational complexity of homomorphic encryption, or sensor fusion limitations for LiDAR-only perception are not addressed.
4. The paper does not compare SPINE against existing privacy-preserving EAI approaches (e.g., federated learning for navigation, differential privacy in perception) in terms of trade-off efficiency, scalability, or usability. Without such comparisons, it is difficult to assess whether SPINE offers meaningful advantages over incremental improvements to existing methods.
5. While the case study quantifies utility (success rate, SPL), it lacks rigorous metrics for measuring privacy preservation (e.g., information leakage rate, re-identification risk, compliance score). The paper’s reliance on qualitative observations (e.g., "visual anonymization masks identity markers") limits the ability to objectively validate SPINE’s effectiveness in mitigating privacy risks.
6. In social EAI scenarios (e.g., care robots, office assistants), privacy risks often involve interpersonal trust and context-dependent social norms (e.g., bystander consent, cultural differences in privacy expectations). The paper’s classification matrix focuses primarily on spatial and data sensitivity, with insufficient attention to these social dimensions of privacy.

**Support:**

3

---

> ### Author Rebuttal · Authors · 2026-03-30
>
> We sincerely thanks for your valuable comments. We hope our response point-by-point below addresses your concerns.
>
> **Q1 More EAI tasks (manipulation, social interaction) or other privacy leakage channels (audio data, trajectory patterns) are expected.**
>
> **Response**: We argue that navigation is a deliberate choice aligned with the ICML Position Track's focus on evidence over exhaustive benchmarks.  It stress-tests our core position: privacy interventions propagate across the entire embodied lifecycle. SPINE’s architectural grammar—from instruction sanitization to data retention—is task-agnostic. In the revised manuscript, we will extend discussion in Section 7  to manipulation, social interaction, and non-visual leakage (audio, trajectories), and map SPINE primitives to them.
>
> **Q2 Provide details on how to dynamically adapt privacy levels in real time.**
>
> **Response**: We argue that aligned with the ICML Position Track, we prioritize an evidence-supported paradigm shift over specific algorithms. SPINE’s L1–L4 matrix provides the formal grammar to navigate lifecycle-wide privacy-utility trade-offs. We will clarify in Section 7 that while real-time orchestration remains future work, SPINE establishes the essential architectural prerequisite for such systems.
>
> **Q3.3. Discuss the feasibility and overhead of L4.**
>
> **Response**: We clarify that aligned with the ICML Position Track, we prioritize an evidence-supported argument over technical benchmarks. Thus, L4 measures (e.g., hardware cutoffs, FHE) are selective safeguards for extreme privacy, not default requirements for low-cost platforms. SPINE provides the necessary architectural grammar to determine when such high overhead is justifiable.
>
> **Q4 Why not compare SPINE against existing baselines?**
>
> **Response**: Aligned with the ICML Position Track, we prioritize an evidence-supported argument over benchmarking against stage-specific baselines (e.g., FedVLN [1], FedVLA[2]). Direct comparison with these baselines is inappropriate as they are fragmented, stage-local patches. Unlike isolated interventions, SPINE provides a lifecycle-wide architectural grammar to prevent compositional leakage that single-module improvements fail to capture. Thus, our goal is not to outperform stage-specific methods but to show their limitation under composition. We will clarify this positioning.
>
> [1] Zhou, K. et al. (2022). FedVLN: Privacy-preserving federated vision-and-language navigation, ECCV-22.
>
> [2] Miao, C. et al. (2025). FedVLA: Federated vision-language-action learning with dual gating mixture-of-experts for robotic manipulation, ICCV-25.
>
> **Q5 Need other privacy metrics; why rely on qualitative observations?**
>
> **Response**: We acknowledge the lack of lifecycle-level metrics and explicitly position this as an open problem. However, aligned with the ICML Position Track, we prioritize architectural paradigms over isolated algorithmic benchmarking. Existing metrics evaluate stage-local patches; a compositional "lifecycle leakage metric" for EAI does not yet exist.  To provide quantitative grounding, we use pixelation scale $K$ as a quantitative privacy proxy. Sensitivity experiments ($K \in \{1, 4, 8, 16, 32\}$) reveal non-linear utility degradation (SR/SPL), mapping operational boundaries:
>
>  | K | Seen SR | Seen SPL | Unseen SR | Unseen SPL |
> |:---|:---|:---|:---|:---|
> | **1 (Vanilla)** | **0.660** | **0.600** | **0.510** | **0.430** |
> | 4 (New) | 0.628 | 0.541 | 0.587 | 0.495 |
> | **8 (Paper)** | **0.460** | **0.340** | **0.360** | **0.270** |
> | 16 (New) | 0.396 | 0.257 | 0.377 | 0.258 |
> | 32 (New) | 0.392 | 0.258 | 0.340 | 0.237 |
>
> Section 4 will clarify $K$ acts as an empirical proxy validating non-linear lifecycle trade-offs. Developing formal compliance metrics will be designated as a future priority within SPINE in Section 7 (Future Directions) of the revised manuscript.
>
> **Q6 Insufficient attention to social dimensions of privacy.**
>
> **Response**: Aligned with the ICML Position Track, we prioritize an evidence-supported architectural argument over exhaustive sociological studies. While social dimensions are important, they remain difficult to formalize due to high subjectivity. Attempting to standardize these variables now would obscure our core message: EAI requires lifecycle-wide architectural coordination. We view SPINE as the foundational infrastructure required before social norms can be enforced. Once this backbone is established, social signals like bystander presence can integrate as high-level triggers for our L1–L4 states. In the revised manuscript, we will expand discussion and position social signals as triggers within SPINE in Section 7 (Future Directions).

---

> > ### Author Rebuttal · Reviewer_2Gug · 2026-04-06
> >
> > The authors addressed most of my concerns, so I keep the original positive rating.

---

### Official Review · Reviewer_pTTZ · 2026-03-11

**Significance:** 2
**Argument Clarity:** 3
**Rating:** 2
**Confidence:** 3

**Questions:**

1. In Table 1, are there guidelines for how to classify each EAI scenario with its tier / level (beyond just learning how to intuit it by seeing examples)?
2. Would it be possible to explain why the privacy-utility trade-off is “non-linear”?
3. How exactly does Figure 2 quantify the privacy-utility trade-off? I would think that to quantify the privacy-utility trade-off you would first need to quantify and precisely define privacy, and then vary the level of privacy and look at the resulting utility.

**Alternative Views Section:**

Yes

**Compliance With Llm Reviewing Policy A Conservative:**

Affirmed.

**Discussion Potential:**

2

**Final Justification:**

I appreciated the depth of the authors' rebuttal, but am keeping my score since I still feel that the paper doesn't quite manage to present a clear argument and supporting evidence.

**Paper Summary:**

This paper argues that component-level interventions are insufficient to achieve privacy in Embodied AI (EAI), and instead calls for a unified framework that enforces privacy requirements across the EAI stages of instruction, perception, planning, and interaction. The authors propose a solution called SPINE that decomposes the EAI pipeline into stages and incorporates a multi-criterion privacy classification matrix that categorizes embodied tasks into different tiers.

**Position:**

Yes

**Position In Title:**

Yes

**Related Work:**

3

**Strengths And Weaknesses:**

Strengths:
* I thought the figures were exceptionally well-done: informative, very nice to look at, and a helpful tool to understand the text.
* The paper is written confidently and I think nicely highlights the authors’ expertise.
* The case study validates SPINE and provides empirical evidence for the hypotheses proposed by the authors.
﻿
Weaknesses:
* This paper read to me like a white paper that identifies a problem and proposes a technical solution. This type of paper has its time and place, but my understanding is that ICML position papers are supposed to stimulate discussion and not necessarily propose a detailed framework. I did feel that there was a position (that privacy needs to be incorporated across the EAI lifecycle rather than patched into isolated stages), but that the paper spends most of its time developing a solution to the identified problem rather than arguing its position with evidence.
* The writing was pretty technical and I felt like the position sometimes got lost in the jargon (at least for me, coming in without an Embodied AI background).
* The paper’s position centers around the idea of a “privacy-utility” trade-off, yet I didn’t really feel like there was a cohesive notion of either privacy or utility. I felt like this was a missed opportunity to showcase the advantages of using the L1-L4 taxonomy since I would think that organizing embodied tasks based on criteria would allow for being able to more easily identify particular privacy definitions or utility metrics. Table 1 kind of hints at this (especially the “privacy-aware techniques” column) but I don’t think it goes all the way.

**Support:**

2

---

> ### Author Rebuttal · Authors · 2026-03-30
>
> We sincerely thanks for your valuable comments. We hope our response point-by-point below addresses your concerns.
>
> **Q1 Position obscured by framework-centric technical details.**
>
> **Response**: Thanks for your comments. We argue that aligned with the ICML Position Track, our paper advances an evidence-based position: EAI privacy should be treated as a lifecycle-level architectural constraint rather than a set of stage-specific patches. SPINE is not a polished technical system, but a concrete vehicle to operationalize this position and expose the non-linear privacy-utility trade-off. In the revision, we will shorten Section 4 (Case Study) so the position is more prominent.
>
> **Q2 The writing was pretty technical.**
>
> **Response**: We agree that domain-specific jargon at times obscured our core position for non-EAI specialists. Our goal, consistent with the ICML Position Track, is to present a timely argument on ML privacy that is accessible to the broader ML community. In the revised manuscript, we will simplify concepts such as "non-linear trade-offs" with narrative examples and add a glossary in the Appendix.
>
> **Q3 Give privacy/utility definitions under L1–L4 criteria.**
>
> **Response**: We agree that our SPINE needs to explicitly define the "how" regarding operational boundaries. We will add this in the revised mansucript, which could truns Table 1 from illustrative to operational.
> Specifically, we explicitly define each level (L1–L4) using a unified tuple $P_L = \{S, I, C, \Phi\}$—where $S$ is Scenario, $I$ is Information Flow, $C$ is Control Primitive, and $\Phi$ is Utility Objective—we turn these representative anchors into formal states. Here is the condensed operational logic for L1–L4, bridged by the pixelation strength ($K$) used in our case study.
>
> * **L1: Public (Utility-Maximizing):** Unrestricted data flow ($I \in \text{Cloud}$) leveraging maximum reasoning power. Logic: $K=0$ (Vanilla baseline); Privacy Cost $\approx 0$.
> $P_{L1} \rightarrow \lbrace S_{pub}, I_{cloud}, C_{null}, \Phi_{max}\rbrace$
>
> * **L2: Internal (Identity-Anonymized):** Hybrid flow removing biometric identifiers ($B$) while preserving spatial geometry. Logic: $K$ is minimal; filters faces/plates but keeps navigation cues intact.
> $P_{L2} \rightarrow \lbrace S_{shared}, I_{hybrid}, C_{f(x)-B}, \Phi_{efficiency}\rbrace$
>
> * **L3: Confidential (Intent-Decoupled):** Local processing to sanitize semantic entities and enforce privacy-aware detours. Logic: $K > 0$; aggressive pixelation reduces visual fidelity to protect environmental context.
> $P_{L3} \rightarrow \lbrace S_{private}, I_{local}, C_{sanitise}, \Phi_{safety\_{first}}\rbrace$
>
> * **L4: Restricted (Zero-Knowledge/Isolation):** Stateless, verifiable isolation mode using non-reconstructable modalities (e.g., LiDAR). Logic: $K \rightarrow \text{Saturation}$; camera cutoff or task prohibition.
> $P_{L4} \rightarrow \lbrace S_{sensitive}, I_{isolate}, C_{FHE/TEE}, \Phi_{min\_{viable}}\rbrace$
>
> **Q4 Provide guidelines to classify EAI scenarios in Table 1.**
>
> **Response**: We agree that the L1–L4 taxonomy should move beyond intuitive examples toward a systematic rule. We will therefore adopt a "Highest-Triggering-Criterion" rule, where a scenario’s level is determined by the most sensitive constraint across the EAI lifecycle. Detailed definitions are provided in Response to Q2.3.
>
> **Q5 Why the privacy-utility trade-off is “non-linear”?**
>
> **Response**: "Non-linear" means utility loss is not proportional to privacy gains. Our navigation study using pixelation ($k$) as a proxy reveals two regimes:
> - Disruption ($L1 \rightarrow L3$): Utility drops sharply as initial constraints break perception-planning coupling.
> - Essential-only ($L3 \rightarrow L4$): Decline tapers off because the system already operates in a "minimal viable" mode with diminishing marginal impacts.
> In the revised manuscript, we will extend a sensitivity analysis of $k$ to Section 3.4 to visualize these non-linear effects.
>
> **Q6 How Figure 2 quantify the privacy-utility trade-off?**
>
> **Response**: We quantify the trade-off by varying the control primitive $C$ within the tuple $P_L = \{S, I, C, \Phi\}$ (see Response Q2.3). Pixelation strength $K$ serves as the operational privacy proxy. By systematically increasing $K \in \{1, 4, 8, 16, 32\}$ across L1–L4, we measure resulting utility via SR and SPL in the table below. The results quantifies the non-linear utility degradation of privacy constraints. In the revised manuscript, we will clarify that pixelation strength $K$ is a controlled proxy, and report SR/SPL across $K$ to show staged degradation with the table below.
>
> | K | Seen SR | Seen SPL | Unseen SR | Unseen SPL |
> |:---|:---|:---|:---|:---|
> | **1 (Vanilla)** | **0.660** | **0.600** | **0.510** | **0.430** |
> | 4 (New) | 0.628 | 0.541 | 0.587 | 0.495 |
> | **8 (Paper)** | **0.460** | **0.340** | **0.360** | **0.270** |
> | 16 (New) | 0.396 | 0.257 | 0.377 | 0.258 |
> | 32 (New) | 0.392 | 0.258 | 0.340 | 0.237 |

---

> > ### Author Rebuttal · Reviewer_pTTZ · 2026-04-03
> >
> > I thank the authors for their rebuttal and sincerely appreciate the detailed responses with added definitions and results. Even so, I will keep my score as is since I still feel that the paper doesn't quite manage to present a clear argument and supporting evidence.

---

### Official Review · Reviewer_dVe5 · 2026-03-13

**Significance:** 3
**Argument Clarity:** 2
**Rating:** 3
**Confidence:** 4

**Questions:**

See Weaknesses.

**Alternative Views Section:**

Yes

**Compliance With Llm Reviewing Policy A Conservative:**

Affirmed.

**Discussion Potential:**

3

**Final Justification:**

I appreciate the author for addressing some of my concerns, but considering that the definition needs further refinement and more evidence is required to support the position, I will keep the score unchanged.

**Paper Summary:**

This paper focuses on the systemic privacy risks of embodied AI (EAI) deployment in sensitive scenarios, pointing out the fundamental flaws of current single-stage, fragmented privacy patches. Therefore, it proposes the SPINE unified privacy framework, which uses privacy as a dynamic control signal throughout the entire EAI lifecycle, and is equipped with an L1-L4 four-level privacy classification matrix for scenario-specific adaptation. The feasibility of the framework is verified through simulation and real-world navigation experiments. This paper quantifies the impact of privacy constraints on EAI performance and defines the operational boundaries of the privacy-utility trade-off.

**Position:**

Yes

**Position In Title:**

Yes

**Related Work:**

2

**Strengths And Weaknesses:**

Strengths:

1. This paper systematically proposes an architectural design approach that uses privacy as a dynamic control signal throughout the entire lifecycle.

2. The constructed four-level privacy classification matrix achieves three-dimensional linkage between scenarios, compliance, and technology, providing differentiated and feasible privacy configuration solutions for EAI applications with different risk levels.

Weaknesses:

1. Scenarios are not specifically defined but described using examples, failing to grasp the core differences.

2. The paper only completed the initial verification of the SPINE framework through embodied navigation tasks, without conducting full-process systematic testing in core high-sensitivity L4 scenarios such as medical companionship and elderly care. The framework's universality and robustness in complex scenarios lack sufficient experimental support.

3. For heavyweight privacy technologies required for L3-L4 high-sensitivity scenarios, such as fully homomorphic encryption and zero-knowledge proofs, the paper does not analyze key issues such as computational latency and hardware overhead in real-time EAI interaction scenarios, indicating significant shortcomings in practical feasibility.

4. Removing landmarks during navigation tasks may lead to privacy leaks, and empirical methods cannot completely guarantee the removal of private information.

**Support:**

3

---

> ### Author Rebuttal · Authors · 2026-03-30
>
> We sincerely thanks for your valuable comments. We hope our response point-by-point below addresses your concerns.
>
> **Q1 Need scenario definitions.**
>
> **Response**: While Table 1 provides the "where," we agree that our SPINE needs to explicitly define the "how" regarding operational boundaries. We will add this in the revised mansucript, which could truns Table 1 from illustrative to operational.
> Specifically, we explicitly define each level (L1–L4) using a unified tuple $P_L = \{S, I, C, \Phi\}$—where $S$ is Scenario, $I$ is Information Flow, $C$ is Control Primitive, and $\Phi$ is Utility Objective—we turn these representative anchors into formal states. Here is the condensed operational logic for L1–L4, bridged by the pixelation strength ($K$) used in our case study:
>
> * **L1: Public (Utility-Maximizing):** Unrestricted data flow ($I \in \text{Cloud}$) to leverage maximum reasoning power. Logic: $K=0$ (Vanilla baseline); Privacy Cost $\approx 0$.
> $P_{L1} \rightarrow \lbrace S_{pub}, I_{cloud}, C_{null}, \Phi_{max}\rbrace$
>
> * **L2: Internal (Identity-Anonymized):** Hybrid flow focused on removing biometric identifiers ($B$) while preserving spatial geometry. Logic: $K$ is minimal; filters faces/plates but keeps navigation cues intact.
> $P_{L2} \rightarrow \lbrace S_{shared}, I_{hybrid}, C_{f(x)-B}, \Phi_{efficiency}\rbrace$
>
> * **L3: Confidential (Intent-Decoupled):** Local processing to sanitize semantic entities and enforce privacy-aware detours. Logic: $K > 0$; aggressive pixelation reduces visual fidelity to protect environmental context.
> $P_{L3} \rightarrow \lbrace S_{private}, I_{local}, C_{sanitise}, \Phi_{safety\_first}\rbrace$
>
> * **L4: Restricted (Zero-Knowledge/Isolation):** Stateless, verifiable isolation mode using non-reconstructable modalities (e.g., LiDAR). Logic: $K \rightarrow \text{Saturation}$; camera cutoff or task prohibition to ensure "what happens in the room, stays in the moment".
> $P_{L4} \rightarrow \lbrace S_{sensitive}, I_{isolate}, C_{FHE/TEE}, \Phi_{min\_viable}\rbrace$
>
> **Q2 Validation on navigation only and needs experiments in complex L4 scenarios.**
>
> **Response**: We intentionally use navigation as a canonical proof-of-concept because it is one of the few tasks that inherently couples the entire "instruction-to-interaction" EAI loop. This allows us to provide deep, legible evidence of how privacy constraints propagate across stages—the core of our position—without the confounding variables of domain-specific manipulation.
>
> More importantly, aligned with the ICML Position track, our goal is to advocate for a lifecycle-level architectural shift rather than to present an exhaustive technical benchmark suite. SPINE provides the "architectural grammar" for high-sensitivity settings, while the navigation study serves as the falsifiable vehicle for that position.
> We agree L4 domains (e.g., healthcare) are critical. We will add a description in Section 1 (Introduction) that navigation is a controlled probe, and explicitly add high-sensitivity domains as primary future validation targets in Section 8 (Conclusion).
>
> **Q3 Analyze practical feasibility of FHE and ZKP for L3-L4 regarding latency and overhead.**
>
> **Response**: We agree this was underdeveloped. While SPINE grounded the cloud-latency vs. edge-resource tension (Section 5) and TEE co-design (Section 7), our analysis of specific heavyweight primitives (FHE/ZKP) was not sufficiently systematic. In the revision, we will extend a concrete analysis of latency, memory, and power overhead (Section 5) , and clarify that such primitives are selectively triggered in L4, not default.
>
> **Q4 Insufficiency landmark removal in preventing residual privacy leakage.**
>
> **Response**: We agree that component-level sanitization is an insufficient standalone privacy guarantee. Our navigation study intentionally used simple pixelation to demonstrate exactly why "empirical patches" fail.
>
> As acknowledged in the submission, even without visible landmarks, privacy risks persist through residual trajectories, semantic maps, and execution logs. This vulnerability precisely motivates SPINE’s position: privacy must be a lifecycle-level architectural constraint. Consequently, SPINE moves beyond visual masking to implement coordinated controls over instruction rewriting, planning traces, and data retention.

---

> > ### Author Rebuttal · Reviewer_dVe5 · 2026-04-07
> >
> > I appreciate the author for addressing some of my concerns, but considering that the definition needs further refinement and more evidence is required to support the position, I will keep the score unchanged.

---

### Decision · Program_Chairs · 2026-04-30

**Decision:**

Accept (regular)

**Comment:**

The argument that the composition of isolated, optimized stages in EAI yields a brittle and privacy-violating whole is insightful, timely, and well-articulated. The proposal to treat privacy as a control signal across the lifecycle is a thoughtful and unifying direction for the community.

I agree that the paper would be strengthened by a more exhaustive threat model, concrete feasibility analysis of heavyweight cryptographic primitives, and a fully automated mechanism for real-time level transitions. However, I judge that these are requirements for a mature technical contribution or a systems paper, not prerequisites for a thought-provoking position paper. I also agree that the paper indeed spends considerable time on SPINE. But I assess that this is not a case of a solution masquerading as a position; rather, the solution is the evidence for the position. The authors use SPINE to show that when one attempts to enforce lifecycle-level constraints, one encounters non-linear trade-offs and regime shifts that are invisible to stage-local thinking. I think all reviewers acknowledged the significance and timeliness of the problem. Their critiques, though valid, largely demand a level of empirical and technical closure that the ICML Position Paper track may not require.